# Single-Grain Gate-All-Around Si Nanowire FET Using Low-Thermal-Budget Processes for Monolithic Three-Dimensional Integrated Circuits

**DOI:** 10.3390/mi11080741

**Published:** 2020-07-30

**Authors:** Tung-Ying Hsieh, Ping-Yi Hsieh, Chih-Chao Yang, Chang-Hong Shen, Jia-Min Shieh, Wen-Kuan Yeh, Meng-Chyi Wu

**Affiliations:** 1Institute of Electronics Engineering, National Tsing Hua University, Hsinchu 30013, Taiwan; tim.hsieh@latticesemi.com (T.-Y.H.); mcwu@ee.nthu.edu.tw (M.-C.W.); 2National Applied Research Laboratories, 3F, No. 106, Ho Ping E. Rd., Sec. 2, Taipei City 10622, Taiwan; 3Taiwan Semiconductor Research Institute, No. 26, Prosperity Road 1, Hsinchu 30013, Taiwan; ping-yi.hsieh@imec.be (P.-Y.H.); chshen@narlabs.org.tw (C.-H.S.); jmshieh@narlabs.org.tw (J.-M.S.); 1305023@narlabs.org.tw (W.-K.Y.)

**Keywords:** monolithic 3D, gate-all-around, nanowire FET, low-thermal budget, location-controlled-grain, laser crystallization, laser activation, laser-assisted salicidation, low power consumption

## Abstract

We introduce a single-grain gate-all-around (GAA) Si nanowire (NW) FET using the location-controlled-grain technique and several innovative low-thermal budget processes, including green nanosecond laser crystallization, far-infrared laser annealing, and hybrid laser-assisted salicidation, that keep the substrate temperature (T_sub_) lower than 400 °C for monolithic three-dimensional integrated circuits (3D-ICs). The detailed process verification of a low-defect GAA nanowire and electrical characteristics were investigated in this article. The GAA Si NW FETs, which were intentionally fabricated within the controlled Si grain, exhibit a steeper subthreshold swing (S.S.) of about 65 mV/dec., higher driving currents of 327 µA/µm (n-type) and 297 µA/µm (p-type) @ V_th_ ± 0.8 V, and higher I_on_/I_off_ (>10^5^ @|V_d_| = 1 V) and have a narrower electrical property distribution. In addition, the proposed Si NW FETs with a GAA structure were found to be less sensitive to V_th_ roll-off and S.S. degradation compared to the omega(Ω)-gate Si FETs. It enables ultrahigh-density sequentially stackable integrated circuits with superior performance and low power consumption for future mobile and neuromorphic applications.

## 1. Introduction

To catch up to Moore’s law, and to benefit from a faster computing speed and economical chip, there are two approaches [1,2] followed and adopted in the semiconductor industry: one is “more Moore” which means keep downsizing the dimension of the transistor using a new device structure or non-silicon materials; the other is “more than Moore” which introduces a heterogeneous integration concept in the out-of-plane direction. The latter is also known as three-dimensional integrated circuits (3D-ICs). Although through silicon via (TSV) technology [3] is the dominating process in modern 3D-ICs, monolithic 3D-ICs [4,5] have drawn increasing attention in recent years which can offer additional transistor layers using either front end of line (FEOL) or back end of line (BEOL) processes for integrated circuits, and thereby provide several advantages, such as the reduction of latency, flexibility, high connectivity, and low power consumption [6]. 

Low-temperature polycrystalline-silicon (LTPS) thin-film transistors (TFTs) fabricated by excimer laser crystallization (ELC) drew tremendous attention in the 1980s [7,8] for their great potential in realizing three-dimensional integrated circuits (3D-ICs). However, the random grain size distribution, the unpredictable grain boundary, and the narrow process window to prevent damages on the bottom tier become ineluctable issues [9,10]. To date, there has been several researches aiming to develop sequentially 3D stackable techniques using a layer transfer method, namely SmartCut^TM^ with semiconductor-on-insulator wafers, e.g., SOI, GOI, or III-V-OI [11,12]. Although this technique could provide better channel quality, it still needs to face the potential high-temperature annealing process of 3D stackable transistor manufacturing. In addition, the process complexity and costly investment may hinder its widespread development. In contrast, there are several groups trying to modify the recrystallization process to prevent the grain boundary effect and to minimize the number of grain boundaries within the active region of a device. One of them is the μ-Czochralski process that tries to control the grain location so then devices can be manufactured within the grain [13,14,15]. However, the thick and rough channel is not appropriate for modern nano-electronics. The main task is to develop a thin and high-crystallinity channel and further low-thermal budget processes for sequentially 3D stackable devices with high performance and low variability. 

In this article, we adopted a μ-Czochralski process using pulse laser crystallization followed by a chemical mechanical polish (CMP) and surface modification steps to fabricate a thin and high-crystallinity location-controlled grain (LCG) Si nanowire channel for its high positioning accuracy, predictability, and compatibility with our present process. A gate-all-around configuration to detach the Si and buried SiO_2_ for a nano-scale transistor, which is much smaller than the grain size of the Si channel, integrated with several innovative low-thermal budget processes was demonstrated. The single-grain GAA Si NW FETs and processes earn not only high performance and low variability but the extremely scaled NW dimension, which, in turn, enables the application for advanced monolithic 3D-ICs.

## 2. Materials and Methods 

The process flow of the monolithic 3D single-grain gate-all-around Si nanowire FET is depicted in Figure 1. The key of fabricating such a sequentially 3D stackable transistor or further integrated circuit lies in developing low-thermal budget processes, which means the substrate temperature should not be higher than 400 °C to keep the metal interconnect reliable and to be compatible with BEOL processes [16,17]. To prepare a high-crystallinity and LCG Si channel, we use a shaped e-beam direct write lithography system and inductively coupled plasma (ICP) etching to create periodic cooling holes on buried oxide as a grain filter before deposing a conformal and 150-nm-thick amorphous Si (a-Si) film. Afterwards, the a-Si film was recrystallized and phase-transformed to a location-controlled and large-grain polycrystalline Si film by Nd:YAG diode-pumped solid-state (DPSS) continuous wave green nanosecond laser annealing (GNS-LA) (Figure 1a). The wavelength, pulse width, scanning speed, beam size, and power of the laser are λ = 532 nm, 13 ns, 25 cm/s, 2 mm × 40 μm, and 5.5 W, respectively. To remove the surface defects caused by the laser process and to improve the overall channel uniformity, the CMP was introduced to thin the thick and large-grain polycrystalline Si film down to below 50 nm (Figure 1b). Afterward, an NW was preliminary defined by a shaped e-beam direct write lithography system and the ICP etching process. 

A whole chemical process was then proceeded to suspend NW in this article as follows: a. soak into two mixture solutions (NH_4_OH:H_2_O_2_:H_2_O = 1:4:20 and HCl:H_2_O_2_:H_2_O = 1:1:6) at 75 °C for 10 min to remove polymer residues and particles caused by the photoresist stripping process (Figure 1c); b. immerse into pure H_2_O_2_ as a strong oxidizing agent for 3 minutes at 100°C to grow sacrificial oxide on the Si surface (Figure 1d); c. suspend the Si NW by removing the sacrificial oxide using dilute HF solution (HF:H_2_O = 1:10) for 5 min (Figure 1e).

The gate stack was fabricated by 3nm-thick HfO_2_ and 50-nm-thick TaN metal layers using atomic layer deposition (ALD) for its excellent step coverage and self-limiting nature around the NW as high-K gate dielectric and metal electrode (HK/MG), respectively (Figure 1f). The gate length of the Si NW FET was defined by shaped e-beam direct write lithography ICP etching process. The exposed channel region was then doped with P^31+^ or BF_2_
^49+^ at a dosage of 5 × 10^15^ cm^−2^ (10 KeV) for N-type or P-type MOSFETs through self-aligned ion implantation and was activated by a CO_2_ FIR-LA to form a highly-activated, abrupt, and ultra-shallow junction source and drain regions (Figure 1g). The wavelength, power, and substrate temperature of FIR-LA were 10.6 μm, 125 W, and 350 °C, respectively.

Moreover, a two-step hybrid laser-assisted self-aligned silicide (Salicide) was also adopted to replace the conventional high-temperature Salicide process [18,19]. In the process, a 10-nm-thin Ni film was sputtered on the NW surface after SiN_x_ spacer formation. The first-step annealing was carried out at 250 °C for 30 s in a rapid thermal annealing (RTA) system to form a high-resistivity Ni_2_Si phase. After removing the unreacted Ni by soaking into a 1:10 HNO_3_ solution at 50 °C for 600 s, the second-step annealing was followed by a CO_2_ FIR-LA with 100W laser power at room temperature to transform the Ni_2_Si phase into a low-resistivity NiSi phase (Figure 1h).

After performing all the low-thermal budget processes to fabricate the single-grain GAA Si NW FETs, SiO_2_ deposition as an interlayer dielectric layer (ILD) followed by a standard metallization process using tungsten (W) metal were then proceeded to accomplish the first stacked device tier (Figure 1i). A two-tier monolithic 3D-IC can be simply realized by repeating the whole low-thermal budget processes (T_sub_ < 400 °C) as we have shown in IEDM 2016 [2]. Figure 2a shows a two-tier monolithic 3D-IC having a metal interconnect to connect the top and bottom Si FETs. By using the proposed single-grain GAA Si NW FETs, the monolithic 3DIC will have better system performance, as illustrated in Figure 2b.

## 3. Results and Discussion

### 3.1. High-Crystallinity and Controlled-Grain Si Film

Figure 3a depicts an SEM image of a random grain growth Si channel on the buried oxide after GNS-LA. Apparently, without any specific structure of the surface or crystallization method, the grain size and grain boundary were randomly distributed. The grain boundary in the Si channel may have unreacted dangling bonds or hydrogen-passivated –Si-H bonds that decrease the carrier mobility and degrade the device performance [20,21,22]. In contrast, Figure 3b shows an SEM image of a Si channel with a regular Si matrix using the location-controlled-grain technique, as illustrated in the inset. The cooling hole acts as a grain filter for lateral grain growth during the melted and quenched laser recrystallization process. The grain boundaries appeared while the neighbor grains were met together. This can be clearly observed after CMP and SEECO etch as shown in Figure 3c. The GAA Si NW FET was designed and intentionally located within a single grain to avoid overlapping the grain boundaries (Figure 3d) which can guarantee better device performance and less output characteristic variation for the monolithic 3DIC design [23,24]. 

Moreover, a quantitative grain size distribution analysis was conducted by the ImageJ software (Figure 3e). Compared to an approximate grain size around 900 nm extracted from the location-controlled-grain Si film, the Si channel without periodic cooling holes exhibits a broad grain size distribution with a standard deviation of 228.3 nm and an average grain size of 631.5 nm from the statistical data in the Figure 3e. The smallest grain size was only about 100 nm that was slightly larger than the size of the GAA Si NW FET. This raises the probability of nano-devices located on the grain boundary. This result suggests that the introduction of periodic cooling holes on buried oxide before the laser crystallization process can effectively achieve a location-controlled-grain Si film. 

In addition to the μ-Czochralski process, the CMP process was also adopted for fabricating a thin and uniform Si channel after the short-pulse GNS-LA. As a result, the CMP not only eliminates the surface roughness but also removes defect and nanocrystalline Si phase layer on the uneven channel surface. Figure 3f shows the AFM images and Raman spectroscopy. The surface roughness of the Si channel dropped from 14.6 to 1.17 nm and the crystallinity was improved while the channel thickness was decreased from 150 to 20 nm. Evidentially, a Raman peak at 520 cm^−1^ representing a nanocrystalline Si signal vanished after the CMP process, revealing the poor crystallinity on the surface due to the slower cooling rate in the interfacial region [25]. 

### 3.2. Gate-All-Around Si Nanowire FET Fabrication

Ultra-thin-body (UTB) silicon in an insulator (SOI) configuration has been widely studied and adopted in our previous works [26,27,28,29] for its outstanding electrostatics due to excellent suppression of the short channel effects (SCEs). However, it is still hard to thoroughly remove the interface traps and fixed oxide defects in back oxide which inevitably degrade the device performance [30,31]. For the recrystallized poly-Si channel, this issue becomes more serious, because the fast quench speed causes more interface defects between the poly-Si and buried oxide interface. Here, a configuration evolution from UTB to GAA was achieved by proposing a whole chemical process to suspend the single-grain Si NW and to eliminate the interface traps and fixed oxide defects. 

By precisely controlling the soaking time of dilute HF that consumes sacrificial oxide and buried oxide, we can simply obtain a single-grain GAA Si NW FET. Figure 4a shows a tilted SEM image of a single-grain GAA Si NW FET with a multi-nanowire channel in order to increase the total driving current. The active region of the FET was intentionally designed and fabricated within a single Si grain. The number of nanowires depends on the grain size and the limitation of the lithographic process. For example, we may put an 8 × 4 static random-access memory (SRAM) mini-array within a grain with a size of 1.04 × 0.83 µm following the 7-nm-node IC design rule [32,33]. For the transistor beyond a 5-nm-node, we may put more FETs or even functional units within a Si grain.

A progressive evolution of device configuration fabricated by increasing HF soaking time is exhibited in Figure 4b–d. An Ω-gate Si FET was demonstrated while the buried oxide underneath was partially etched away, as shown in Figure 4c. The coverage of the HK/MG stack can be increased to enhance controllability by a longer soaking time to totally detach the buried oxide that touches the Si channel. The ultimate GAA Si NW FET with W**_fin_**/H**_fin_**/L**_gate_** = 15 nm/15 nm/30 nm was achieved by 5 min soaking time as presented in Figure 4d. The high-resolution transmission electron microscopy (HRTEM) image in Figure 4e exhibits an ALD-deposited HfO_2_ and a single-grain Si lattice. The selected area diffraction pattern (SADP) for the laser-crystallized channel in Figure 4f reveals that the Si channel is highly recrystallized.

### 3.3. Highly Activated Ultra-Shallow-Junction Formed by FIR-LA

One of the bottle-necks in monolithic 3D GAA Si NW FET devices is to form highly activated (low sheet resistance R_sh_) source/drain extensions with abrupt, ultra-shallow junctions (USJ) [34,35]. How to repair the amorphized surface bombarded by dopant species as well as to avoid severe dopant diffusion that causes the SCEs during the high-energy activation and heat dissipation through the ILD downward to the bottom metal interconnect and device/circuits become a key challenge.

The introduced FIR-LA is an intraband excitation process [36,37]. It can selectively transfer energies to the free carriers and repairs defects or activates the dopants without damaging the top HK/MG nano-structure of a gate-first device. The limited heating zone and short process duration result in fast heat dissipation and therefore keep the bottom metal interconnects and the device/circuit “cool”.

A high-resolution transmission electron microscope (HRTEM) image of a Si after the dopant implantation process is shown in Figure 5a and clearly indicates that the surface is bombarded into disorder. The amorphous Si layer is about 15-nm-thin, appearing after the P^31+^ dopant implantation at a heavy dosage of 5 × 10^15^ cm^−2^ and 5 KeV ion energy. With an FIR-LA condition of 125 W laser power for 100 µs at 350 °C annealing temperature, the amorphous layer shall be epitaxial regrown. Figure 5b shows the dopant profiles with and without the FIR-LA using a secondary ion mass spectrometer (SIMS) and they are compared to conventional rapid thermal annealing (RTA). Apparently, a 37.6 nm diffusion depth at the level of 10^18^ dopant concentration was observed in conventional RTA at 800 °C for 10 s annealing time which increases the risk of current leakage between source and drain due to lateral diffusion in the nanodevices. In contrast, the FIR-LA contributes only a 7.3 nm diffusion depth which reflects the fact that a short-pulse and long-wavelength laser annealing effectively suppresses the SCEs. The series resistance in the inset table, which is another index of the annealing process and dominates the final performance of a scaling device, was further reduced from 213 to 93 Ohm/γ by adopting the FIR-LA to replace conventional RTA. The nearly diffusionless dopant profiles and low resistivity represent the advantage of such a low-thermal budget FIR-LA in monolithic 3D-ICs. 

### 3.4. Hybrid Laser-Assisted Salicidation

Nickel salicidation is a crucial process for advanced MOS transistors for its low Si consumption, low formation temperature, and low line width sheet resistance dependence as compared to cobalt or titanium silicide [18,19]. Previous studies proved that a rough NiSi film and NiSi_2_ facet (111) formed during conventional two-step RTA, which will lead to Fermi-level pining and a junction spiking leakage current [38,39]. Additionally, the second step of conventional two-step RTA usually reaches 400 to 600 °C to form a low-resistivity NiSi film, which is slightly higher than the BEOL CMOS process requirement that may cause a reliability issue for monolithic stacking [9,10]. Here, we introduce a hybrid laser-assisted salicidation process, that is first annealed at 250 °C for 30 s in an RTA system followed by second-step FIR-LA with 100 µs process time at room temperature (T_sub_ = 25 °C), to form a thin and uniform Si-rich NiSi_x_ film as well as to suppress the formation of NiSi_2_ facet (111).

The X-ray diffraction (XRD) patterns of nickel silicide formed by the hybrid laser-assisted salicidation process were displayed in Figure 6a. Five nickel monosilicide (NiSi) peaks, corresponding to the (0 0 2), (1 1 1), (2 0 2), (1 0 3), and (0 1 3) lattice planes, were observed and no NiSi_2_ facet (1 1 1) was found, which promises a high-quality and low-resistivity NiSi film. The morphology of the NiSi film prepared by the hybrid laser-assisted salicidation process was analyzed by high-resolution transmission electron microscopy (HRTEM). The second-step FIR-LA melts the high-resistivity Ni_2_Si phase near the surface and the melt front diffuses downward to the Si and Ni_2_Si interface. The mixing of Ni_2_Si and Si occurs via liquid phase diffusion [38], which leads to a uniform NiSi film without heterogeneous aggregation and spike at the interface between the NiSi and the Si, which is confirmed in Figure 6b,c [40]. The atomic percentage of Si, identified by energy-dispersive X-ray spectroscopy (EDS), slightly raises from 49.87 (point 1) to 54.74% (point 3), implying Ni_2_Si diffused from the surface to the bottom.

### 3.5. Device Uniformity Characterization and FIR-LA Validation for Monolithic Three Dimension Integrated Circuits Application

Figure 7a provides the transfer characteristics at |V_d_| = 1 V of the Ω-gate Si FETs and the GAA Si NW FETs with W_fin_/H_fin_/L_gate_ = 15 nm/15 nm/30 nm simultaneously using location-controlled-grain technique. The Ω-gate Si FETs exhibits subthreshold swing (S.S.~71 mV/dec.), driving current 291 µA/µm (n-type) and 274 µA/µm (p-type) at V_th_ ± 0.8 V. The single-grain GAA Si NW FETs have better gate controllability and therefor exhibits superior electrical properties with steeper subthreshold swing (S.S.) of about 65 mV/dec., higher driving currents (Ion) of 327 µA/µm (n-type) and 297 µA/µm (p-type) at V_th_ ± 0.8 V, and higher on/off current ratio (I_on_/I_off_) of >10^5^. The off currents of the GAA Si NW FETs, extracted from the curves at V_g_ = 0 V, reach about 1nA/µm which was much lower than the Ω-gate Si FETs. 

To extract the threshold voltage (V_th_) and S.S. dependencies of these two types of single-grain transistor on the gate length (L_g_), the device structures having a similar size were chosen for comparison. The data were collected and plotted in Figure 7b. According to the result, the V_th_ decrease while the L_g_ becomes shorter. Further, the S.S. is found to be increased with the shrinkage in L_g_. It indicates that the SCEs dominate the output characteristics at sub-100 nm of both transistors. The slower V_th_ roll-off and S.S. ramp up rate in the line graph of the GAA Si NW FET elucidate that the gate-all-around configuration has better ability in suppressing SCEs. This evidence can be found in Figure 7c as well. The S.S. rises from 65 to 121 mV/dec. (ΔS.S. = 56 mV/dec.), while the W_Fin_ increases from 10 to 50 nm for the Ω-gate Si FETs. Comparatively, the GAA Si NW FETs reveals a lower S.S. in the same W_Fin_ and a smaller ΔS.S. (43 mV/dec.).

By introducing the LCG technique for monolithic 3D-ICs, we can prevent the electrical property variation caused by the random grain size and the unpredictable boundaries. Figure 7d,e show the cumulative plots of S.S. and V_th_ of the GAA Si NW FETs with and without the LCG technique within an 8-inch Si wafer. Both plots demonstrate the single-grain GAA Si NW FETs with the LCG technique have smaller S.S. (from 19.2 to 14.7 mV/dec at FWHM) and V_th_ deviations (from 0.51 to 0.44 V at FWHM). 

To validate the feasibility of the monolithic 3D-IC sequential integration process, the transfer characteristics of a GAA Si NW FET with a two-layer metal interconnect (M1-M2) were prepared and tested before and after FIR-LA. Evidently, the lack of obvious changes in the I_d_-V_g_ behaviors before and after the FIR-LA in Figure 7f ensures the compatibility of the monolithic 3D-IC sequential integration process.

## 4. Conclusions

In summary, novel methodologies to fabricate GAA Si NW FETs with high performance and low power consumption were reported. By integrating the location-controlled-grain technique, laser crystallization, far-infrared laser dopant activation, and hybrid laser-assisted salicidation, sub-50 nm GAA Si NW FETs with lower sensitivity to Vth roll-off and subthreshold swing degradation as the gate length scaled down were presented. Furthermore, electrical validation results provide convincing evidence for the feasibility of monolithic three-dimensional integrated circuits. The advanced 3D architecture demonstrated in this article enables high-bandwidth sequentially stackable circuits to achieve superior performance and low power consumption for future mobile and neuromorphic applications.

## Figures and Tables

**Figure 1 micromachines-11-00741-f001:**
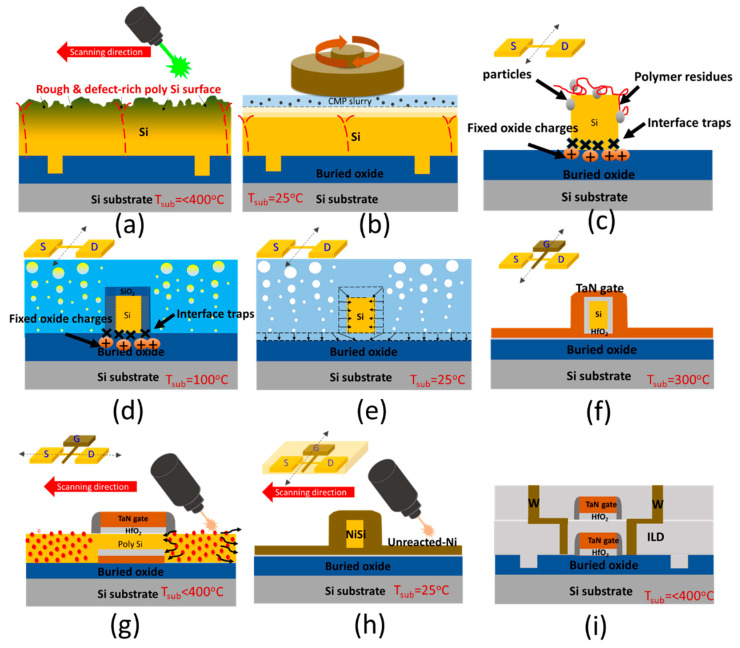
Schematics of process steps for 3D stackable single-grain gate-all-around (GAA) Si nanowire (NW) FET: (**a**) a-Si film was crystallized by GNS-LA; (**b**) the chemical mechanical polishing (CMP); (**c**) NW with defects was preliminary defined by lithography and ICP etching process in a Si gain; (**d**) soak into two mixture solutions to remove polymer residues and particles and immerse into pure H_2_O_2_ to form sacrificial oxide on the Si NW surface; (**e**) suspend the Si NW by removing the sacrificial oxide using dilute HF solution; (**f**) high-K/metal gate (HK/MG) stack fabricated by ALD; (**g**) implantation and far-infrared laser annealing (FIR-LA) for dopant activation, (**h**) hybrid laser-assisted self-aligned silicide and (**i**) interlayer dielectric layer (ILD) deposition and following standard metallization using tungsten (W) interconnect.

**Figure 2 micromachines-11-00741-f002:**
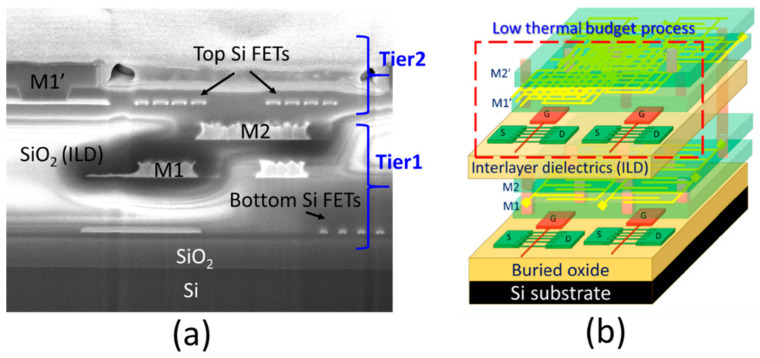
(**a**) FIB image of a 3D sequential integration with two stacking tiers and metal interconnects. (**b**) Schematic illustration of a monolithic three-dimensional integrated circuit (3DIC) using single-grain GAA Si NW FETs.

**Figure 3 micromachines-11-00741-f003:**
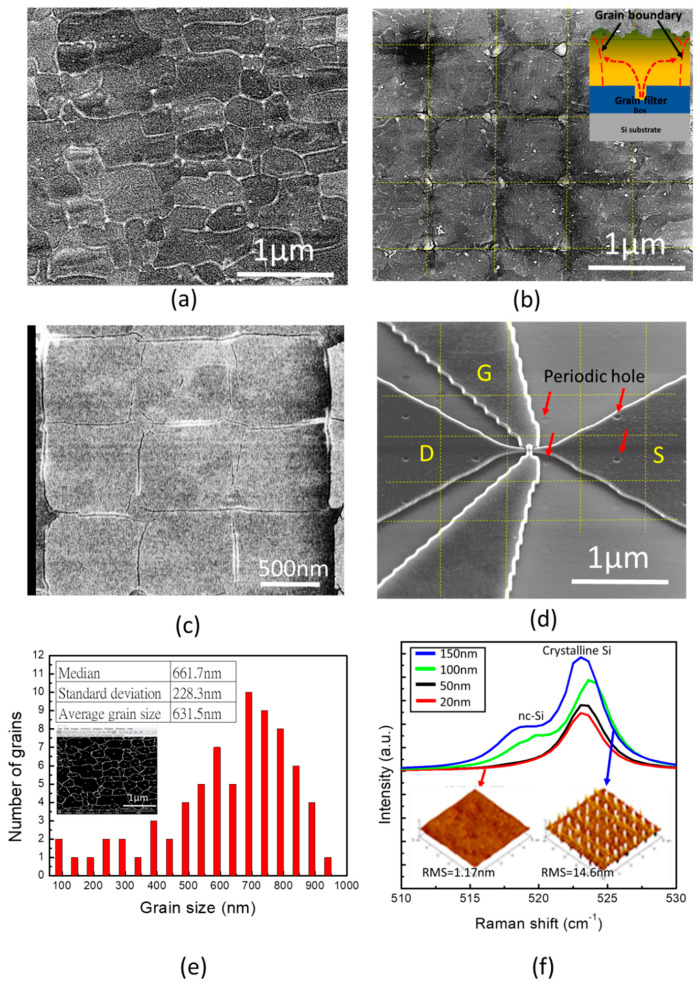
(**a**) SEM image of random grain growth poly-Si channel; (**b**) SEM image of a Si matrix using the LCG technique with the grain filter illustrated in the inset; (**c**) SEM image of a thin LCG Si film after CMP and SEECO etch (solution of K_2_Cr_2_O_7_ water mixed with HF) to reveal the grain boundaries; (**d**) SEM image of a single grain GAA Si NW FET intentionally located in a Si gain; (**e**) grain size distribution of a Si channel without the LCG technique and the inset is the image analysis conducted by the ImageJ software. (**f**) The Raman spectrums of the Si channel before and after the CMP thinning process and inset AFM images show the related surface morphology and the root mean square roughness.

**Figure 4 micromachines-11-00741-f004:**
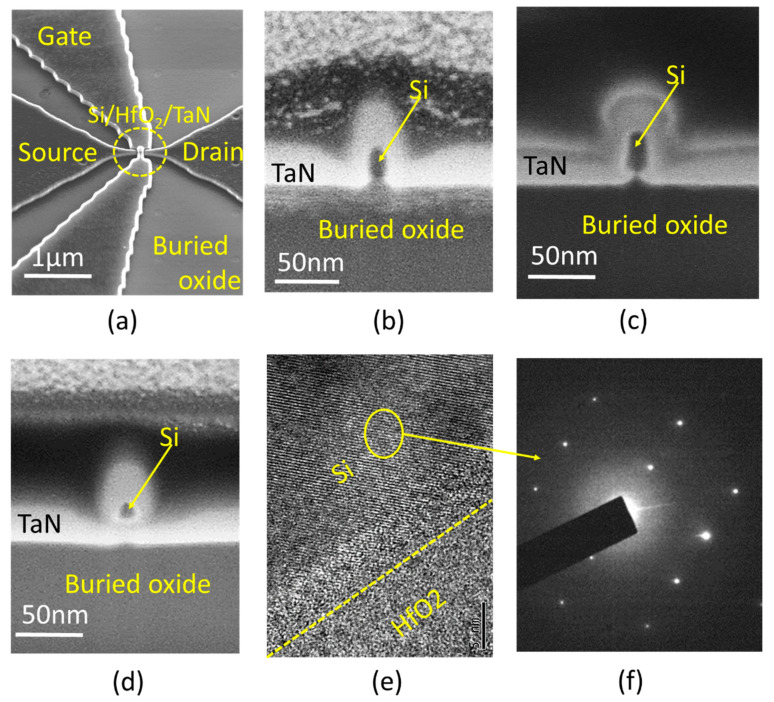
(**a**) A tilted SEM image of a multi-wire GAA Si NW FET. The configurations of (**b**) tri-gate, (**c**) Ω-gate, and (**d**) GAA Si FET with different soaking times in the channel fabrication step. (**e**) A HRTEM image of single-grain Si and HfO_2_ dialectic layer. (**f**) A selected area diffraction pattern (SADP) from Si.

**Figure 5 micromachines-11-00741-f005:**
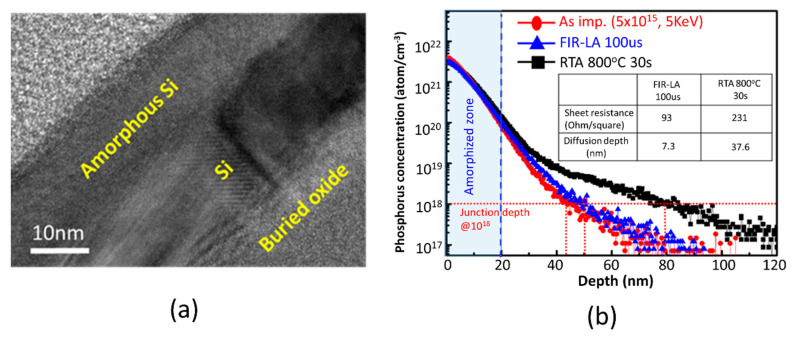
(**a**) A HRTEM image of Si after dopant implantation. (**b**) Dopant profiles characterized by a secondary ion mass spectrometer (SIMS). The inset table presents the sheet resistance differences between FIR-LA and RTA.

**Figure 6 micromachines-11-00741-f006:**
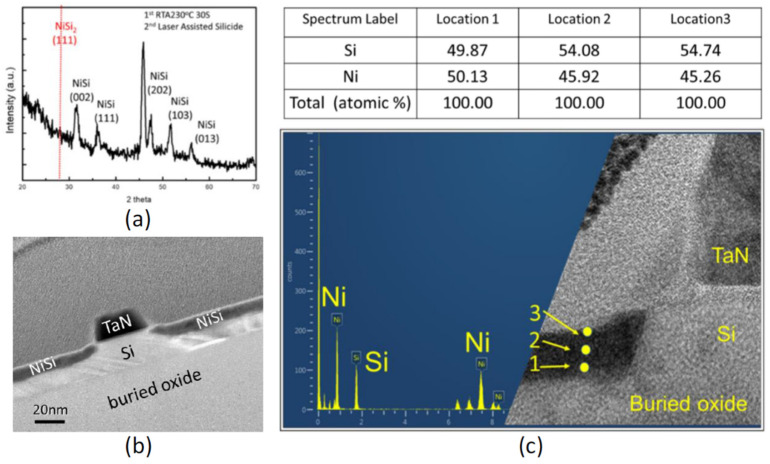
(**a**) The XRD patterns of the nickel silicide formed by the hybrid laser-assisted salicidation; (**b**) a TEM image of a Si FET with a flat and uniform NiSi film in the source and drain region; (**c**) energy-dispersive X-ray spectroscopy (EDS) information of the NiSi film in the source and drain region at various depths.

**Figure 7 micromachines-11-00741-f007:**
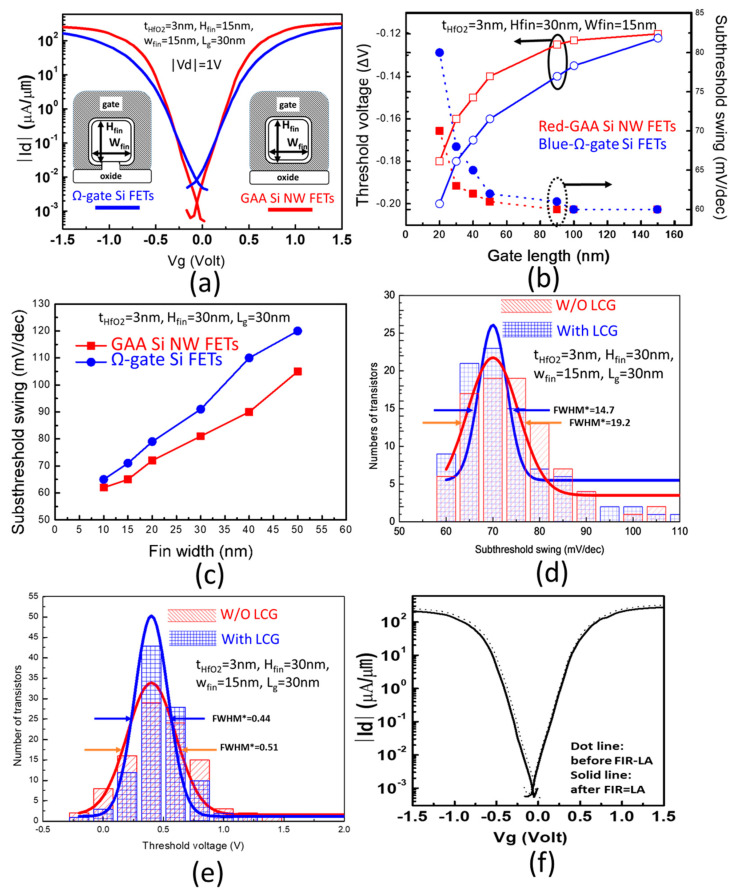
(**a**) A comparison of transfer characteristics between Ω-gate Si FETs and GAA Si NW FETs; (**b**) the dependencies of threshold voltage and S.S. on the transistor gate length (L_g_); (**c**) the dependency of S.S. on the fin width (W_Fin_); the distributions of (**d**) S.S. and (**e**) V_th_ of the GAA Si NW FETs with and without the LCG technique; (**f**) the transfer characteristics of a GAA Si NW FET before and after the FIR-LA.

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
