# Peer review of "Single-Grain Gate-All-Around Si Nanowire FET Using Low-Thermal-Budget Processes for Monolithic Three-Dimensional Integrated Circuits"

_micromachines, 2020, doi:10.3390/mi11080741_

Round 1

Reviewer 1 Report

This study has demonstrated gate-all-around Si nanowire TFET by utilizing laser annealing process for various process steps such as Si crystallization, dopant activation, and source/drain salicidation. Laser processes introduced this study cover whole of the fabrication processes of Si transistors, and the discussion is well-organized systematically. Therefore, the achievements of this study will contribute to technological developments for monolithic three-dimensional integrated circuits under a low thermal budget. On the other hand, I have some comments to improve the manuscript outlined below.

(1) In the introduction part, please make the originality of this study clearer. For example, some papers, which cite reference # 14 [ Ishihara et al., Thin Solid Films 427, 77 (2003). R. Ishihara et al., IEEE Trans. Electron Dev. 51, 500 (2004). C.-C. Tsai et al., Appl. Phys. Lett. 91, 201903 (2007).], and reference # 20 already discuss transistor performances fabricated by utilizing position-controlled quasi-single-crystal Si layers.

(2) In Fig. 1, the position and the direction of the cutting plane are unclear, especially for Figs. (g) and (h).

(3) Why the cross-sectional structure of Si shown in Fig. 4(b), 4(c), and 4(d) are so different after long HF soaking? The size of Si nanowire may change the electrical characteristics largely and, hence, it is hard to distinguish impacts between the size of the Si nanowire size and coverage of the gate stack, as discussed in Fig. 6.

(4) In Fig.5, how does TEM image after FIR-LA look? Also, how much concentration of the impurity is activated, or how much is the activation ratio?

(5) Variation of carrier mobilities is one of the biggest concerns for the transistors fabricated by utilizing Si single-grain. Also, is there any correlation between variations of Vth, SS, or mobility and location of the device in Si single grain (ie. distance from the periodic hole or grain boundary)?

(6) Minor errors,

i) Line 100: BF249+ should be BF249+ (“2” should be subscript)

ii) Line 183~: Fig. 3 should be Fig. 4.

iii) Line 206: A region where the high-concentration impurity is implanted should not be a Si “channel”.

Reviewer 2 Report

The authors present the full fabrication process flow of GAA Si NWFET by using a low thermal budget process. Their novel technique of  using shaped e-beam direct write lithography  and relative ICP etching allowed them to create periodic cooling holes. By implementing this approach, the authors can effectively achieve a location-controlled-grain Si film which has a positive impact on Si NWFET performance.

Adding to this, laser crystallization, far-infrared-laser dopant activation and hybrid laser-assisted salicidation, led to sub-50 nm GAA Si NW FET with lower sensitivity to Vth roll-off and subthreshold-swing degradation as the gate length is scaled down, compared to other geometries like omega gate electrode.

In overall, the current study is well organized and nicely presented. I recommend the publication of this paper in your journal, after the implementation of the following points:

  1. In the abstract section, rephrasing is needed. A better summary of the findings is expected making clear to the reader the key output of the study. Besides, English syntax should be improved.
  2. In lines 54 and 58 the authors mentioned about Si, however they should add the word nanowire, which is the subject of the paper
  3.  in lines 72 and 73 the authors should insert corresponding abbreviations for HK/MG and FIR-LA
  4. In line 97, more technical information is needed to explain how is possible to form an all around the NW the gate dielectric and gate electrode. Previously the authors mentioned that they suspend the Si NWs by removing the oxide all around the NW. But it is not clearly explained how they place the dielectric and the gate electrode below the NW. Is it possible only by tuning the etching process duration to achieve this? they come back at this point in line 172, but this process should be clear to the reader at the very beginning of the manuscript.
  5. In line 106, the authors should insert a corresponding reference for the silicide process
  6. In line 113, the authors should define what M1,M2 is (they do this only in line 283
  7. In line 126, the author should define what SEECO is
  8.  In 251, the authors should correct the type "dimention" with "dimension" . besides in this figure, in panel b, its better to use open symbols for the SS (right axis) and closed symbols for the Vth (left axis)
  9. In line 256, the authors mention fin's width but they are not describing what is this exactly. is it equivalent with NW diameter?
  10. In line 261, the authors are giving corresponding values only for GAA devices and not for omega gate structures. they should enter values for omega devices therefore. the same is true for abstract section.
  11. In 271, there is typing error in output characteristics (the authors insert out, instead output)
  12. it will be better to update the reference list with more recent papers on corresponding state of the art devices. For example: Japanese Journal of Applied Physics 59, 070908 (2020). Besides, there are some published papers in the past by the authors, but they are not citing them. For example the paper entilted:"Enabling Low Power BEOL Compatible monolithic 3D+ nanoelectronics for IoTs Using
    Local and Selective Far-Infrared Ray Laser Anneal Technology" in IEEE 2016, and "Footprint-efficient and power-saving monolithic IoT 3D+ IC constructed by BEOL-compatible sub-10nm high aspect ratio (AR>7) single-grained Si FinFETs with record high Ion of 0.38 mA/μm and steep-swing of 65 mV/dec. and Ion/Ioff ratio of 8".
    There are many similirities of the present study with these two aforementioned papers, therefore the authors should explicity describe the novely of the present article compared to them and also the state of the art.
